# Anti-Ebola virus mAb 3A6 protects highly viremic animals from fatal outcome via binding GP$_{(1,2)}$ in a position elevated from the virion membrane

Kathryn M. Hastie [1], Zhe Li Salie [1,18,21], Zunlong Ke[2,3,15,21], Peter J. Halfmann [4], Lisa Evans DeWald[5], Sara McArdle [6], Ariadna Grinyó [7,20], Edgar Davidson[7], Sharon L. Schendel [1], Chitra Hariharan[1], Michael J. Norris [1,19], Xiaoying Yu [1,17], Chakravarthy Chennareddy[8], Xiaoli Xiong [2,16], Megan Heinrich [9], Michael R. Holbrook [5], Benjamin Doranz [7], Ian Crozier [10], Yoshihiro Kawaoka [4,11,12,13], Luis M. Branco[9], Jens H. Kuhn [5] ✉, John A. G. Briggs [2,3] ✉, Gabriella Worwa[5] ✉, Carl W. Davis[8] ✉, Rafi Ahmed [8] ✉ & Erica Ollmann Saphire [1,14] ✉

Monoclonal antibodies (mAbs) against Ebola virus (EBOV) glycoprotein (GP$_{1,2}$) are the standard of care for Ebola virus disease (EVD). Anti-GP$_{1,2}$ mAbs targeting the stalk and membrane proximal external region (MPER) potently neutralize EBOV in vitro and are protective in a mouse model of EVD. However, their neutralization mechanism is poorly understood because they target a GP$_{1,2}$ epitope that has evaded structural characterization. Using X-ray crystallography and cryo-electron tomography of mAb 3A6 complexed with its stalk–MPER epitope, we reveal a previously undescribed mechanism in which 3A6 binds to a conformation of GP$_{1,2}$ that is lifted from the virion membrane. We further show that in both domestic guinea pig and rhesus monkey EVD models, 3A6 provides therapeutic benefit at high-viremia advanced disease stages and at the lowest dose yet demonstrated for any anti-EBOV mAb-based monotherapy. The findings reported here can guide design of next-generation highly potent anti-EBOV therapeutics and vaccines.

Ebola virus (EBOV; family *Filoviridae*: species *Orthoebolavirus zairense*) causes severe and frequently fatal acute human disease in outbreaks that can result in thousands of deaths. Complicating containment efforts, EBOV may persist subclinically in survivors for years and reignite outbreaks. Ebola virus disease (EVD) can be prevented with two licensed vaccines and treated with approved monoclonal antibody (mAb) therapeutics[1]. However, even with approved mAbs, outcomes remain poor in patients with high viral loads and/or advanced disease, and the incidence of viral persistence is unknown. As such,

identification and optimization of novel mAbs are needed to address these gaps.

All advanced anti-EBOV mAb-based therapeutics and vaccines target protein spikes protruding from virion envelopes[2]. Each spike comprises a single EBOV-encoded glycoprotein (GP$_{1,2}$), synthesized by translation of a preprotein that is cleaved in the Golgi apparatus into GP$_1$ and GP$_2$ subunits. A disulfide bond links these two subunits to form heterodimers (Fig. 1a top) that assemble into GP$_{1,2}$ trimers[3,4]. GP$_1$ contains a heavily glycosylated mucin-like domain

A full list of affiliations appears at the end of the paper. ✉e-mail: kuhnjens@niaid.nih.gov; briggs@biochem.mpg.de; gabriella.worwa@nih.gov; cwdavi2@emory.edus; rahmed@emory.edu; erica@lji.org

**Fig. 1 | Structures of human mAb 3A6 in complex with the Ebola virus glycoprotein stalk–MPER peptide. a** Top: Schematic of proteolytically processed mature Ebola virus (EBOV) glycoprotein (GP$_{1,2}$) using the amino acid residue numbering of its uncleaved precursor minus signal peptide residues. Middle and bottom: EBOV GP$_{1,2}$ constructs used in this study. Inset: Sequence alignment of glycoprotein subunit 2 (GP$_2$) stalk–MPER amino-acid sequences. Aligned are the stalk–MPER transition areas (with the two regions separated by a vertical black line) of all six known orthoebolaviruses. The predicted linear epitope of 3A6[11] is indicated by a purple box. The EBOV residues observed to interact with 3A6 in the crystal structure are highlighted in dark orange and the corresponding regions in glycoproteins of the other orthoebolaviruses are highlighted in light orange. Orthoebolaviruses associated with fatal human disease are in bold type. **b** Top and front view of 3A6 (grey) bound to the EBOV stalk–MPER peptide (orange). The

heavy chains (HCs) and light chains (LCs) of 3A6 are highlighted in dark and light grey, respectively. Light-chain complementarity determining regions (CDRs) are colored in greens while heavy-chain CDRs are colored in purples. **c** The stalk region of the peptide epitope forms extensive interactions with CDR-H3. **d** The majority of contacts in MPER region of the peptide epitope are through CDR-L1 and -L3. **e** Electrostatic and (**f**) hydrophobic surfaces of 3A6 are shown, with residues contributing to each surface indicated. Changes to either D632 and P636 are associated with viral escape from 3A6. BDBV Bundibugyo virus, BOMV, Bombali virus, HR heptad repeat regions, IFL internal fusion loop, IFR interfacial region, mAb, monoclonal antibody, MLD mucin-like domain, MPER membrane proximal external region, RESTV Reston virus, SUDV Sudan virus, TAFV Taï Forest virus, TM transmembrane [domain].

(MLD) that obscures the upper and outer portions of GP$_{1,2}$ and a glycan cap domain that shields the virion receptor-binding site in the GP$_1$ core from the host immune response (Fig. 1a top). Upon virion entry, host cell cathepsins proteolytically process GP$_{1,2}$ in the endosome to remove the MLD and glycan cap domain to expose the GP$_1$ core and binding site for virion receptor NPC intracellular cholesterol transporter 1 (NPC1)[5–7]. GP$_2$, a typical class I fusogen, mediates fusion of virion and endosomal membranes to release viral

ribonucleocomplexes into the target cell[8]. GP$_2$ contains an internal fusion loop (IFL), two consecutive heptad repeat regions (HR1 and HR2), a membrane proximal external region (MPER), and a C-terminal transmembrane (TM) domain (Fig. 1a top). HR2 also called the stalk, is largely alpha-helical and connects the GP$_2$ core to the MPER and TM domain[7].

Major recognition sites for anti-EBOV mAbs are the GP$_1$ MLD and glycan cap domain; the GP$_{1,2}$ trimer base; and the GP$_2$ IFL, stalk, and

MPER[9–17]. Stalk−MPER is of special interest for therapeutic and vaccine design because of the potent neutralization activity demonstrated by mAbs that bind to it. Indeed, the region has 70% sequence conservation among all six orthoebolaviruses and, for the three viruses that can cause fatal disease, Bundibugyo virus (BDBV), EBOV, and Sudan virus (SUDV), the sequence conservation increases to ≈90%[9,11,13]. Despite these important features, mAbs against the stalk−MPER is the least well-characterized of anti-orthoebolavirus mAbs in part because of a comparative lack of structural information about this region; studies to date have used GP$_{1,2}$ constructs in which MPER was deleted (for increased solubility and stability). Only one structure, that of the BDBV stalk−MPER targeted by mAb BDBV223, has been examined[7]. Curiously, the angle at which BDBV223 binds to the BDBV GP$_2$ epitope suggests that, in current models of the EBOV GP$_{1,2}$ structure, access would be occluded by the viral or host cell membrane[18]. As such, the mechanism by which mAbs that target the stalk−MPER access their epitope remains unclear, further questioning whether access is associated with therapeutic efficacy for patients infected with EBOV or related viruses.

In this work, we use x-ray crystallography to illuminate the discrete binding footprint of mAb 9.6.3A6, a highly potent stalk-MPER-targeted mAb. Using cryo-electron tomographic analysis, we show that this mAb gains access to its occluded epitope by stabilizing EBOV GP$_{1,2}$ in a lifted state. Finally, we demonstrate complete mAb 9.6.3A6-mediated post-exposure protection in the stringent guinea pig and rhesus monkey models of EVD.

## Results

### Structure of mAb 3A6 in complex with the Ebola virus stalk−MPER peptide

mAb 9.6.3A6 (henceforth abbreviated as "3A6 IgG") was isolated from a human survivor of the 2013−16 Western African EVD outbreak 6 months after hospital discharge[11]. The predicted linear epitope of 3A6 IgG encompasses GP$_{1,2}$ residues 626−640 and extends from the C-terminal end of the stalk to the start of the MPER[11] (Fig. 1a, bottom/inset, orange box, outlined in purple in the sequence alignment). To determine the mode of molecular recognition of EBOV by 3A6 IgG, we crystallized the 3A6 antigen-binding fragment ("Fab", henceforth abbreviated as "3A6") alone (Supplementary Fig. 1) and in complex with a 14-amino-acid peptide having a sequence corresponding to the EBOV GP$_{1,2}$ stalk−MPER epitope (residues 626−640; Fig. 1b−f). Crystals of 3A6 diffracted to 2.5 Å and had an asymmetric unit containing four Fab molecules. Meanwhile, crystals of the 3A6-stalk−MPER peptide complex diffracted to 1.27 Å and had one Fab fragment in the asymmetric unit (Supplementary Table 1). The 3A6 structure was essentially identical in the unbound and peptide-bound states as evidenced by the 0.46 Å root-mean-square deviation (RMSD) over the entirety of the paratope (Supplementary Fig. 1b). Residues I627−G639 of the 3A6-peptide epitope are visible (Fig. 1c−f, Supplementary Fig. 1a−e). The EBOV stalk−MPER peptide is α-helical from its N terminus (I627) to residue T634 and then slightly unravels through the visible terminus at residue G639 (Supplementary Fig. 1).

The paratope of 3A6 can roughly be divided into stalk-binding or MPER-binding areas. The N-terminal/stalk portion of the peptide (residues 627−631) is enveloped by complementarity determining regions (CDRs) H1, H3, and L2 (Fig. 1b, Supplementary Fig. 1). CDR-H3 residues R98, S100, T101, and Y104 form extensive hydrogen bonds to GP$_2$ residues H628, D629, and D632 (Fig. 1c, e; Supplementary Fig. 1), CDR-H1 residue E31 contacts MPER residue K633, and Y54 of CDR-L2 interacts with surrounding water molecules to contact MPER residue T634 (Fig. 1c; Supplementary Fig. 1). Heavy-chain residues Y32 and Y104 and light-chain residues Y54, L58, and F99 engage in hydrophobic interactions with this region of the peptide (Fig. 1f).

The MPER/C-terminal portion of the peptide is surrounded by CDR-H2, CDR-H1, and CDR-L3. Antibody residues H31 and S32 (CDR-L1), T97

(CDR-L3), and N59 (CDR-H2) engage MPER residues L635, P636, D637, and Q638 (Fig. 1d, e). Residue M33 of the heavy chain and A96, F99, and L101 of the light chain form a hydrophobic pocket around P636 (Fig. 1f). MPER residues D632 and P636 are particularly crucial and form extensive interactions with the Fab (Fig. 1c−f).

### Comparison to stalk-binding mAb BDBV223

Only one other high-resolution structure of an orthoebolavirus GP$_2$ stalk−MPER-targeted mAb is documented. The structure of mAb BDBV223 was determined alone and in complex with its peptide epitope from BDBV[18]. Comparison of the mode of recognition of 3A6 and BDBV223 for their cognate targets reveals several differences. BDBV223 binds GP$_2$ residues 620−634, placing its epitope primarily within HR2 (residues 599−632) (Supplementary Fig. 2a). In addition to an upward shift, the manner in which BDBV223 binds to its epitope is markedly different from that of 3A6. In contrast to 3A6, for which the peptide binds directly in the groove between the heavy and light chains (Fig. 1b), the heavy chain of BDBV223 forms the bulk of interactions with its peptide epitope, with only minor contributions from the light chain of CDR-L3 (Supplementary Fig. 2b). Moreover, while there is no induced fit for 3A6 (Supplementary Fig. 1b), binding of BDBV223 to its peptide epitope results in substantial rearrangements in the heavy-chain CDRs as compared to the unbound Fab. Furthermore, 3A6 binds to EBOV GP$_{1,2}$ at an ≈80° angle, while BDBV223 binds at an ≈117° angle (Supplementary Fig. 1c).

Despite these differences, there are curious similarities between 3A6 and BDBV223. Both 3A6 and BDBV223 are escaped by changes at aspartate (D632 and D624, respectively) or proline (P636 or P634, respectively) residues (Fig. 2 and King et al., 2019[18]). Although located in different regions of the GP$_2$ stalk−MPER, each of these residues interacts with the respective mAb in a similar manner. In particular, both D624 and D632 of GP$_2$ are coordinated by CDR-H3 residues and both P634 and P636 of GP$_2$ occupy a hydrophobic pocket within CDR-L3 (Fig. 1c−f and Supplementary Fig. 2b).

### EBOV GP$_{1,2}$ MPER residues D632 and P636 are critical for mAb 3A6 binding

3A6 IgG binds to and neutralizes EBOV but not SUDV in vitro[11]. In the 3A6 IgG peptide epitope that includes stalk−MPER residues I627-G639, four residues differ substantially between EBOV and SUDV GP$_2$: K vs. N at 633, T vs. P at 634, D vs. N at 637, and G vs. D at 639 (Fig. 1a, bottom/inset). Using a cell-based antibody-binding assay, we compared binding of 3A6 IgG to full-length (MLD-containing) EBOV GP$_{1,2}$ with each of these four residues changed individually or in combination with the corresponding SUDV residues (Fig. 2a and Supplementary Fig. 2a); EBOV GP$_{1,2}$ lacking residues 312−462 of the MLD and residues 644−676 of the TM domain (GP$_{1,2ΔTM/ΔMLD}$) was created. In addition, we used an enzyme-linked immunosorbent assay (ELISA) to evaluate 3A6 binding to recombinant EBOV GP$_{1,2ΔTM/ΔMLD}$ ectodomains containing the same amino acid changes (Fig. 2b). In either case, none of the individual amino-acid residue changes affected 3A6 IgG binding, but binding was completely inhibited when all four residues were changed to the SUDV counterparts (Fig. 2, Supplementary Fig. 3a). Binding of control mAbs that target quaternary epitopes in the fusion domain of GP$_2$ was not impacted, indicating the overall conformation of GP$_{1,2}$ was not altered by any change(s) (Supplementary Fig. 3b, c).

Next, we used alanine-scanning mutagenesis of GP$_{1,2}$ to identify individual residues throughout the epitope that are crucial for 3A6 binding. We individually replaced alanine residues (wild-type [WT] Ala to Ser) at positions 627−639 of EBOV GP$_{1,2}$ lacking the MLD (GP$_{1,2ΔMLD}$; Fig. 1a) and analyzed each resulting EBOV GP$_{1,2ΔMLD}$ for 3A6 reactivity by flow cytometry (Supplementary Table 2, Supplementary Fig. 4). Notably, both D632A and P636A showed reduced 3A6 IgG binding by more than 80% relative to WT GP$_{1,2ΔMLD}$ (Fig. 2c). Meanwhile, neither

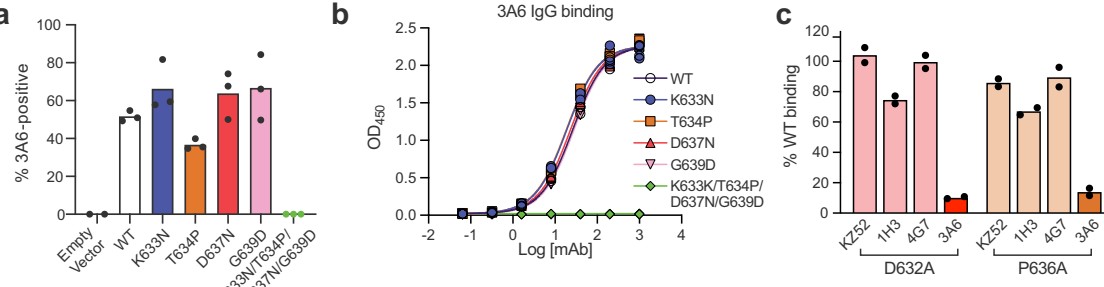

**Fig. 2 | Residues D632 and P636 of the Ebola virus glycoprotein MPER are key for mAb 3A6 binding. a** Cells expressing wild-type (WT) Ebola virus (EBOV) glycoprotein (GP$_{1,2}$) or recombinant EBOV GP$_{1,2}$ with the indicated changes to cognate amino-acid residues found in Sudan virus (SUDV) glycoprotein subunit 2 (GP$_2$) were incubated with 3A6 immunoglobulin G (IgG) and then stained with DyLight 488 anti-human IgG for detection by fluorescence microscopy, followed by quantification of antibody-positive cells. **b** Enzyme-linked immunosorbent assay (ELISA) binding curves for 3A6 IgG to purified EBOV GP$_{1,2\Delta TM/\Delta MLD}$ or variants thereof containing the indicated amino-acid residue changes (symbols represent $n = 3$ technical replicates). **c** Flow cytometry analysis of mAb binding to cell-surface expressed EBOV GP$_{1,2}$ bearing a D632A or P636A change. Black dots in (**a**, **c**) indicate the values for individual transfection experiments. mAb monoclonal antibody, MPER membrane proximal external region.

change substantially affected binding of control mAbs KZ52, 1H3, or 4G7 that target conformational epitopes on the base (KZ52 and 4G7) and glycan cap (1H3) of EBOV GP$_{1,2}$[7,19,20] (Fig. 2c). These results are consistent with those of previous studies, in which morphologically authentic "biologically contained" EbolaΔVP30 virions[21], passaged in the presence of 3A6 IgG led to the emergence of glycoproteins bearing P636S and P636Q changes[11]. Therefore, we next evaluated neutralization of P636S-bearing EbolaΔVP30-eGFP virions by multiple mAbs using a plaque-reduction assay. P636S abolished neutralization activity of both 3A6 and 1E6, another stalk-binding mAb[11], but did not affect neutralization of mAbs targeting the glycoprotein core (Supplementary Table 3).

### Binding of mAb 3A6 lifts EBOV GP$_{1,2}$ relative to the membrane surface

All GP$_{1,2}$ structures in which the stalk region can be modeled include a fibritin trimerization motif fused to the GP$_2$ C terminus[22,23]. It is not known whether inclusion of exogenous trimerization domains induces a non-native association of the HR2/MPER helices or if the stalk region adopts a more open and flexible conformation in the context of membrane-bound GP$_{1,2}$. Certainly, superimposition of the 3A6–stalk–MPER structure onto a fibritin-stabilized GP$_{1,2\Delta TM/\Delta MLD}$ trimer (PDB 5JQ7) revealed steric clashes of the bound 3A6 with the other two GP$_2$ monomers of stalk–MPER (Supplementary Fig. 5a). Hence, the tightly bundled conformation of GP$_2$ HR2 observed in this crystal structure is incompatible with Fab binding.

To assess the stoichiometry of the 3A6-EBOV GP$_{1,2\Delta TM/\Delta MLD}$ interaction, we used complementary biophysical techniques to evaluate Fab binding to soluble GP$_{1,2\Delta TM/\Delta MLD}$ ectodomains. Size-exclusion chromatography coupled to multi-angle light scattering (SEC-MALS) revealed that 3A6-GP$_{1,2\Delta TM/\Delta MLD}$ complexes elute as a single peak with a molecular mass consistent with that of a fully occupied GP$_{1,2\Delta TM/\Delta MLD}$ trimer (Supplementary Fig. 5b). Peaks corresponding to binding of one or two 3A6 molecules to GP$_{1,2\Delta TM/\Delta MLD}$ were not observed. Composition-gradient multi-angle light scattering (CG-MALS) analysis supported this result, revealing that three copies of 3A6 can bind to one GP$_{1,2\Delta TM/\Delta MLD}$ trimer and do so with equal affinities ($K_D = 52.15$ [±1.3] nM; Supplementary Fig. 5c). Moreover, negative-stain EM (nsEM) analysis of 3A6–GP$_{1,2\Delta TM/\Delta MLD}$ complexes also showed binding of three 3A6s to the stalk–MPER (Supplementary Fig. 5d).

These analyses conclusively demonstrated that GP$_{1,2\Delta TM/\Delta MLD}$ ectodomains can bind to three 3A6 simultaneously. However, the protein used for these studies did not contain an exogenous trimerization domain, a feature that, if it were there, would have constrained access to the epitope. Similarly, the GP$_2$ C termini in the natural

membrane-anchored form would have less freedom of movement than would be available in the GP$_2$ termini of untethered ectodomains. To image 3A6 bound to GP$_2$ in its natural transmembrane form, we produced filamentous EBOV-like particles (VLPs) consisting of EBOV matrix protein (VP40) and full-length GP$_{1,2}$. VLPs were incubated with 3A6 and analyzed via cryogenic electron tomography (cryo-ET), followed by subtomogram averaging (Fig. 3a, Supplementary Fig. 6, Supplementary Table 4). Tomographic reconstruction of VLP GP$_{1,2}$ in the absence of 3A6 was unsuccessful, perhaps due to flexibility of the GP$_{1,2}$ on the viral surface. Thus, to represent the state of GP$_{1,2}$ in the absence of 3A6, we complexed VLPs with the Fab of the well-characterized mAb KZ52, which binds to the fusion peptide of EBOV GP$_{1,2}$[7]. We also imaged a ternary complex of both 3A6 and KZ52 Fab bound to VLP GP$_{1,2}$ to determine if binding of one influenced the binding of the other or the position of the GP$_{1,2}$ in relation to the VLP membrane.

Reconstructions of VLP GP$_{1,2}$–3A6 (18 Å) and VLP GP$_{1,2}$–KZ52 (8.7 Å) showed the expected chalice-shaped GP$_{1,2}$ trimer with extra densities anchored to the stalk and to the base of the chalice, corresponding to 3A6 and KZ52 Fab, respectively (Fig. 3b, c). Likewise, the ternary complex of GP$_{1,2}$ with both Fabs confirmed that each binds to discrete non-overlapping sites on GP$_{1,2}$ (Fig. 3b, c). The strong Fab density in each reconstruction suggests that the bulk population of GP$_{1,2}$ is fully occupied by the bound Fab(s), however, we have not performed a classification aimed at resolving minority populations with fewer numbers of Fabs. Importantly, the KZ52 epitope at the base of GP$_1$ is sufficiently separated from the stalk–MPER 3A6 binding site, and its binding does not disrupt the native structure of GP$_{1,2}$. Although the overall conformation of GP$_{1,2}$ was unchanged regardless of the Fab(s) bound to it, we did observe marked differences in the position of GP$_{1,2}$ relative to the surface of the VLP. In the presence of 3A6, the GP$_{1,2}$ body was displaced vertically away from the VLP membrane by approximately 3 nm as compared to the position of GP$_{1,2}$ when bound to KZ52 alone (Fig. 3b). We conclude that 3A6 binding induces this vertical displacement since VLP GP$_{1,2}$ complexed with both 3A6 and KZ52 Fabs together resulted in a similarly lifted GP$_{1,2}$ (Fig. 3c) with densities that overlapped with VLP GP$_{1,2}$–3A6 (Fig. 3d).

To gain insight into the location of the 3A6 epitope in the context of virion-displayed GP$_{1,2}$, we used AlphaFold 2[24,25] to model the various states GP$_{1,2}$ may sample. Twenty models were generated (five models generated in each of four independent trials) and curated to remove those in which the TM domain was disordered or not positioned in a manner consistent with membrane association (Supplementary Fig. 7a, b). The remaining ten models were sorted into those that adopted a compact MPER structure ($n = 6$; Supplementary Fig. 7c) and

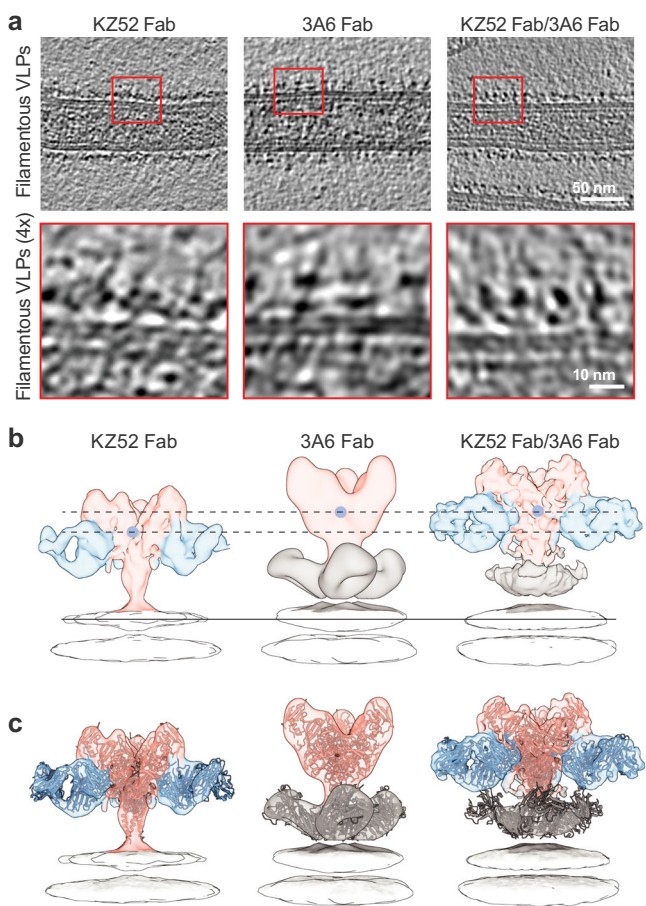

**Fig. 3 | Binding of mAb 3A6 lifts Ebola virus glycoprotein relative to the membrane surface. a** Representative tomographic slices of filamentous Ebola virus (EBOV) virion-like particles (VLPs) bound to the indicated Fabs ($n = 18, 26, 42$ tomograms for VLP $GP_{1,2}$–KZ52, VLP $GP_{1,2}$–3A6, and VLP $GP_{1,2}$–KZ52 + 3A6, respectively). EBOV matrix protein (VP40) of VLPs is visible as a dotted layer underneath the lipid bilayer. The bottom row corresponds to magnified view of areas enclosed by red boxes in the top row. **b** Isosurface representations of sub-tomographic reconstructions of EBOV glycoprotein ($GP_{1,2}$; pink) on the surface of VLPs bound to KZ52 Fab (blue), 3A6 (tan), or both KZ52 Fab and 3A6 aligned along the plane of the outer layer of the VLP membrane. Blue spheres indicate the center of the $GP_{1,2}$ model. **c** Models of $GP_{1,2}$ (PBD code 5JQ3) and Fab (PDB 3CSY and 7RPU) are shown docked into each reconstruction. Density corresponding to the VP40 layer has been removed for clarity in panels b and c. Fab fragment antigen binding.

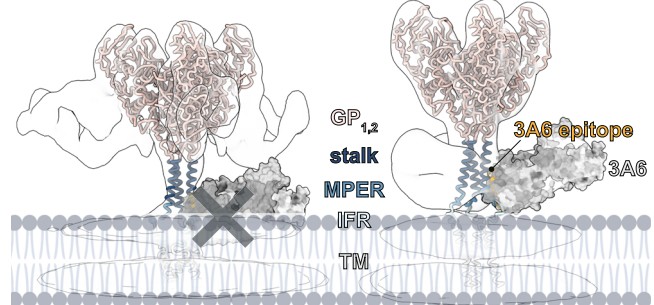

**Fig. 4 | Vertical breathing of the Ebola virus glycoprotein stalk and MPER may allow 3A6 to bind.** AlphaFold 2[25,26] models suggest that the Ebola virus (EBOV) glycoprotein ($GP_{1,2}$) can adopt compact (right) and extended (left) membrane proximal external region (MPER) and interfacial region (IFR) conformations. The compact conformation is incompatible with 3A6 binding. Instead, 3A6 requires MPER to be in an extended conformation. mAb monoclonal antibody, TM transmembrane [domain].

those that adopted an extended MPER ($n = 4$; Supplementary Fig. 7d). The positional arrangement of the MPER and TM domain in the "compact MPER" models was similar to that of the previously reported solution structure of EBOV MPER/TM[8] (Supplementary Fig. 7e), in which the soluble and dynamic N-terminal region (residues 632–640) is linked to the well-ordered TM domain (residues 656–676) through a short interfacia l $\alpha$-helical region and flexible turn (residues 641–655). Further, we found that models with a compact MPER best fit the sub-tomographic reconstruction of $GP_{1,2}$ in complex with KZ52 (Supplementary Fig. 7c, e), while those with an extended MPER fit the reconstruction of $GP_{1,2}$ bound to 3A6 (Supplementary Fig. 7d, f).

We next modeled the location of the 3A6 epitope in a representative of each of the compact and extended MPER models and found that, as expected, the epitope is highly occluded in the compact model and exposed in the extended model (Fig. 4). Notably, we find no evidence that 3A6 interacts with membrane alone (Supplementary Fig. 3a), suggesting that 3A6 does not penetrate the membrane to gain access to its epitope. Instead, we propose that 3A6 binds the epitope when $GP_{1,2}$ is tilted and/or lifting up and retracting down. Once

engaged by 3A6, $GP_{1,2}$ is stabilized in a lifted state. The outwardly located epitope is consistent with the requirement that the stalk must be exposed for 3A6 to bind. It remains unclear whether this position is a natural feature of the $GP_2$ stalk or if it occurs upon 3A6 binding.

### Low-dose mAb 3A6 monotherapy is protective in stringent models of EVD

Previous in vivo studies demonstrate that prophylactic administration of 3A6 IgG protects laboratory mice from fatal outcomes after exposure to a typically lethal dose of mouse-adapted EBOV (100% protection after a 100-μg dose [≈5 mg/kg] and 50% protection at a 25 μg dose [≈1.25 mg/kg])[11]. To determine whether 3A6 is also effective in the post-exposure setting, groups of six (three male and three female) guinea pigs were first exposed intraperitoneally (IP) to 1000 PFU of domesticated guinea-pig-adapted EBOV (Day 0). On Day 3, the guinea pigs were either left untreated or treated IP with a single 5 mg dose of anti-EBOV antibodies 3A6 IgG, 1A2 IgG (targets the EBOV $GP_2$ fusion loop), or 7G7 IgG (targets an unknown epitope on EBOV $GP_{1,2}$[11]) or the anti-influenza A virus (FLUAV) antibody 42-2D2. All animals in the untreated group and those treated with 1A2 or 7G7 succumbed to EBOV infection. All but two of the 42-2D2-treated control animals succumbed to EBOV infection. In contrast, all guinea pigs treated with 3A6 survived and exhibited few or no clinical signs of disease (Fig. 5a, Supplementary Fig. 8a-c).

Guinea pigs offer a more stringent model of EVD compared to mice, but the rhesus monkey model better recapitulates key features of EVD and is generally preferred over rodent models for development of EVD medical countermeasures[1]. As a proof of concept, we randomized four rhesus monkeys into two groups: administered treatment ($n = 3$, rhesus monkeys 1–3) and no treatment ($n = 1$). All monkeys were exposed via intramuscular (IM) injection to a typically lethal 1000-PFU dose of EBOV (Day 0). On Day 4 and Day 7 after exposure, the treatment group monkeys given 25 mg/kg of 3A6 in phosphate-buffered saline (PBS) intravenously, whereas the control monkey was given intravenous PBS only. EBOV replication was confirmed in all monkeys on Day 4 by plaque assay titration and quantitative real-time reverse transcription polymerase chain reaction (RT-qPCR), with $10^4$–$10^6$ EBOV PFU per mL and $10^8$–$10^{10}$ EBOV glycoprotein (*GP*) gene equivalents per mL of serum (Supplementary Fig. 8d). Notably, these high viremic levels could still be reversed by 3A6 administration, as evidenced by a decrease in viral load after the first dose on Day 4 and continued reduction to below the limit of detection by Day 21 (Supplementary Fig. 8d). Clinical signs consistent with EVD, observed in monkeys 1 and 3 as early as Day 4, resolved by Day 13 (Supplementary Fig. 8e). Notably, monkey 3 had significantly elevated AST activity and rapidly

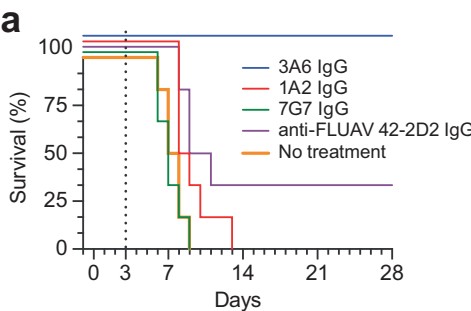

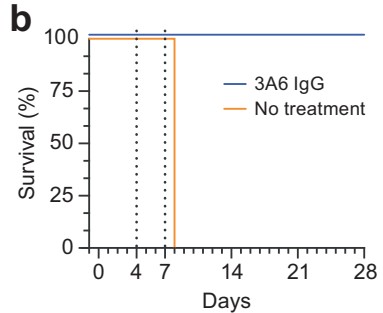

**Fig. 5 | Low-dose mAb 3A6 monotherapy protects domesticated guinea pigs and rhesus monkeys against Ebola virus. a** Domesticated guinea pigs ($n = 6$ per group) were exposed to a typically lethal 1000-PFU dose of domesticated guinea-pig-adapted Ebola virus (EBOV) on Day 0. On Day 3, the indicated mAbs were administered at 5 mg each in phosphate-buffered saline (PBS). Control guinea pigs were either given an influenza A virus (FLUAV)-specific human immunoglobulin G1 or were untreated. **b** Rhesus monkeys ($n = 3$) were exposed to a typically lethal 1000-PFU dose of EBOV on Day 0. On Day 4 and Day 7, 25 mg/kg of 3A6 was administered in PBS. One control monkey was given PBS. Treatment days are indicated by dotted lines. IgG immunoglobulin G, mAb monoclonal antibody.

rising serum creatinine concentrations, suggesting a marked reduction in the glomerular filtration rate (the initial rise was similar to that of the control animal) with pathologic evidence of EBOV-induced liver injury (Supplementary Fig. 9a, b). Nonetheless, this animal still recovered after 3A6 IgG treatment (Fig. 5b, Supplementary Fig. 8e). All three monkeys in the treatment group survived, whereas the control monkey was found dead on Day 8 (Fig. 5b). These preliminary results demonstrate that post-exposure dosing of 3A6 IgG alone reverses the course of EBOV infection and protects against fatal outcomes. Additional studies, including a larger cohort of NHPs and differential dosing (i.e., lower dose of antibody, variation in the number of doses and day of treatment) will further illuminate the therapeutic impact of 3A6 treatment.

## Discussion

Vaccines against EVD are approved[26] but none are in place for infections caused by other filoviruses. Similarly, the human standards-of-care monotherapy ansuvimab (mAb114) and cocktail atoltivimab/maftivimab/odesivimab (REGN-EB3) are also specific for EBOV and are dosed at 50 mg/kg each (150 mg/kg total for three). As such, antibody therapeutics that can be used at a low dose to reverse advanced disease are urgently needed to treat people with filovirus infections who live in countries with limited resources. The EBOV GP$_{1,2}$ stalk–MPER is of interest for therapeutic/vaccine design due to its relatively high amino acid sequence conservation among all orthoebolaviruses, indicating that a single mAb targeting this region could have therapeutic activity against infections by any of these viruses. Moreover, known anti-orthoebolavirus stalk and/or MPER mAbs are highly potent neutralizers in vitro[11,18,27], suggesting that they may be applied in much lower doses compared to mAbs that are currently used in the clinic.

Here we built on previous in vitro and prophylactic laboratory mouse efficacy studies of the EBOV GP$_{1,2}$ stalk–MPER-binding mAb 3A6 by demonstrating complete post-exposure protection in stringent models of EVD in domesticated guinea pigs and rhesus monkeys. 3A6 showed unprecedented potency in the rhesus monkey model of disease. In the pilot study performed here, monkeys presenting with severe disease signs associated with fatal outcomes were treated with a single 25-mg/kg dose of 3A6, which reduced Day 4 viral loads of $10^9$–$10^{10}$ PFU per mL to undetectable levels by Day 21. Concomitantly, 3A6 reversed clinical signs of advanced disease and decreased elevated liver enzyme activities and serum creatinine concentrations to baseline. The administered dose is half that used for mAb114 monotherapy and one-sixth of that used for the REGN-EB3 mAb cocktail in the same animal model (albeit administered on different days)[10,28]. Our data therefore pave the way for development of novel therapeutics that potentially expand the treatment window later in the EVD course for

effective intervention in highly viremic patients. Such therapeutics could increase the likelihood of survival for this group of patients seen relative to currently approved mAb therapeutics.

We previously hypothesized that binding of BDBV223, an anti-stalk antibody targeting a similarly occluded epitope in the EBOV-related Bundibugyo virion[18], requires either bending or lifting of GP$_{1,2}$. In this study, we experimentally addressed this hypothesis using 3A6, which, at the outset appears to bind to an epitope that is even more occluded than that bound by BDBV223. Our structural and modeling studies suggest that the 3A6-targeted epitope is in close apposition to the membrane and perhaps occluded in some conformations. The inherent flexibility of the stalk region may allow EBOV GP$_{1,2}$ to bend or extend, thus enabling 3A6 access and therefore the ability to lift GP$_{1,2}$ from the surface of the membrane.

As high-resolution structures of full-length transmembrane GP$_{1,2}$ are missing, we do not yet know if the three membrane-proximal stalks are bundled or separated in the natural protein. If bundled, the first 3A6 to bind would likely induce some unbundling and displacement. We also do not know if the first 3A6 to bind would induce a tilt, in which one epitope is exposed, and the second and third 3A6 to bind would support a general lift with all epitopes exposed or if the first to bind would be enough to support a generally lifted and exposed conformation for all three instances of the 3A6 epitope on the trimer.

We hypothesize that by binding to the stalk, 3A6 achieves its potent neutralization activity through the blocking of conformational changes needed to drive fusion of the virion and cell membranes. Human immunodeficiency virus 1 GP160 and influenza virus A (FLUAV) hemagglutinin can also be tilted by binding of anti-MPER antibodies[29,30], indicating that positional flexibility is a common property of class I fusogens.

In conclusion, our studies establish 3A6 as a potent immunotherapeutic against EBOV that achieves complete protection against advanced disease at the lowest dose yet observed for a monotherapy via a previously undescribed mechanism of action. The next desired feature of stalk and MPER-targeted mAbs is breadth: 3A6-like antibodies with pan-orthoebolavirus activity likely exist. Such antibodies could be used at even lower concentrations and at more advanced stages across the filovirus disease spectrum. We have shown that binding of stalk and MPER-targeted antibodies requires a flexible HR2 helical bundle that can accommodate antibody binding. Tethers such as exogenous trimerization domains or presentation of the minimal stalk–MPER peptide on protein scaffolds will likely prevent the elicitation of antibodies that can then recognize native full-length GP$_{1,2}$. This study provides new blueprints for the development of stalk–MPER-targeted vaccines that elicit potent and broad immune responses.

## Methods

### Cell lines

Human embryonic kidney (female) epithelial Expi293F cells (Thermo Fisher Scientific, Waltham, MA, USA) were cultured on orbital shakers in Expi293 expression medium (Thermo Fisher Scientific) at 37 °C in a humidified atmosphere containing 8% carbon dioxide ($CO_2$). HEK 293 T cells (American Type Culture Collection [ATCC] Manassas, VA, USA; #CRL-3216, human, female) were cultured in high-glucose Dulbecco's modified Eagle's medium (DMEM) containing L-glutamine (Invitrogen, Carlsbad, CA, USA), supplemented with 10% heat-inactivated fetal bovine serum (FBS; Omega Scientific, Tarzana, CA, USA) and 1% penicillin–streptomycin solution (Thermo Fisher Scientific). Cells were maintained at 37 °C in a humidified atmosphere with 5% $CO_2$. *Drosophila* Schneider 2 (S2) cells (Thermo Fisher Scientific) were cultured with Schneider's *Drosophila* medium (Thermo Fisher Scientific) in stationary flasks at 27 °C. Stable cell lines were adapted to serum-free conditions and maintained on orbital shakers at 27 °C.

### Antibody and antibody fragment expression, purification, crystallization, and visualization

Protein fragment generation, protein and protein fragment purification, crystallization, X-ray structure determination, and negative-stain electron microscopy were performed according to standard protocols.

**Protein expression and purification.** Immunoglobulins were expressed with an Expi293 Expression System (Thermo Fisher Scientific). Light and heavy chain-encoding plasmids were prepared using an endotoxin-free kit (NucleoBond Xtra Midi Plus EF; Takara Bio, San Jose, CA, USA) and used to transfect Expi293 cells at a 2:1 ratio of light chain to heavy chain with Expifectamine 293 transfection reagent (Thermo Fisher Scientific) according to the manufacturer's instructions. mAbs containing supernatants from transfected cells were clarified by centrifugation and then incubated with protein A agarose resin (GenScript, Piscataway, NJ, USA) in batch format overnight, followed by washing, elution, and buffer exchange into Dulbecco's PBS (DPBS; Thermo Fisher Scientific) as previously described by Smith et al., 2009[31]. Antibodies used in vivo were verified to be endotoxin-free using a commercial detection kit (Thermo Fisher Scientific). All antibodies produced in this study were expressed as human IgG1.

Fab fragments were generated from purified IgG1s through digestion with 3% immobilized papain (Thermo Fisher Scientific) for 2 h, followed by purification with a Mono Q anion exchange chromatography column (GE Healthcare, Chicago, Illinois, USA) and size-exclusion chromatography with a Superdex 75 Increase 10/300 GL column (GE Healthcare) in 1X tris(hydroxymethyl)aminomethane-buffered saline (tris-buffered saline [TBS]; Thermo Fisher Scientific). Fractions with pure Fab were concentrated using Ultra Centrifugal Filter Units (Amicon, Miami, FL, USA). Epitope peptide representing $GP_{1,2}$ residues 626–640 was chemically custom-synthesized by Thermo Fisher Scientific and purified via high-performance liquid chromatography (HPLC).

Recombinant WT and Ebola virus/H.sapiens-tc/COD/1976/Yambuku-Mayinga $GP_{1,2}$ ectodomain variants, the latter lacking residues 312–462 of the MLD and residues 644–676 of the transmembrane domain ($GP_{1,2\Delta TM/\Delta MLD}$), were expressed in S2 cells. All constructs carried a C-terminal double-strep tag for affinity purification. Stably transfected cells were selected with 6 μg/mL puromycin (InvivoGen, San Diego, CA, USA). The resulting strep-tagged proteins were purified using a 5-mL StrepTrap column (Cytiva, Marlborough, MA, USA) according to the manufacturer's protocol and then further purified with size-exclusion chromatography (SEC) using a Superdex 200 column (Cytiva) in 1X TBS.

**Protein crystallization.** 3A6 was crystallized in 28% polyethylene glycol (PEG) 400, 0.1 M 4-(2-hydroxyethyl)-1-piperazineethanesulfonic acid (HEPES) + sodium hydroxide (NaOH) pH 7.5 buffer, and 0.2 M calcium chloride ($CaCl_2$) (all constituents from Hampton Research, Viejo, CA, USA) at 20 °C. To form the 3A6–peptide complex, purified 3A6 was concentrated to 5 mg/mL, combined with a five-fold excess of peptide, and incubated at 4 °C for 18 h. The Fab–peptide complex was crystallized in 30% PEG 3000, 0.1 M tris pH 7.0, and 0.2 M sodium chloride (NaCl) (all constituents from Hampton Research) at 20 °C. Crystals were flash-cooled in liquid nitrogen, with 15% ethylene glycol (Hampton Research) as a cryoprotectant.

**X-ray data collection and protein structure determination.** X-ray diffraction data of Fab–peptide complexes were collected on beamline 12-2 at the Stanford Synchrotron Radiation Lightsource, and Fab diffraction data were collected on beamline 23ID-B at the Advanced Photon Source[32,33]. One dataset for the Fab crystal was used, and two datasets from separate Fab–peptide complex crystals were merged for processing using autoPROC with XDS[34,35] for indexing and integration, followed by POINTLESS[36] and AIMLESS[37] programs for data reduction, scaling, merging, and calculation of structure factor amplitudes and intensity statistics. One Fab–peptide complex per asymmetric unit was present in space group P1 21 1 ($a = 52.3$ Å, $b = 66.4$ Å, $c = 68$ Å, $\alpha = \gamma = 90°$, $\beta = 104.2°$), and four Fabs were present in the asymmetric unit of the Fab structure in space group P1 ($a = 53.7$ Å, $b = 65.7$ Å, $c = 125.6$ Å, $\alpha = 98.7°$, $\beta = 91.4°$, $\gamma = 96.0°$). Crystal structures were determined by molecular replacement using PHASER[38] within the CCP4 package[39], using a homology model predicted with the SWISS-MODEL server[40]. Iterative manual model rebuilding was performed using Coot[41] and refined with PHENIX[42]. The peptide was built into different Fourier maps and calculated prior to inclusion of the respective structural elements. Final atomic coordinates and structure factors of the Fab–peptide complex and apo–Fab structures were deposited in the Protein Data Bank (PDB) under identification numbers (IDs) 7RPU and 7RPT, respectively. Figures were created in PyMOL (http://www.pymol.org/) and Chimera X.

**Size-exclusion chromatography coupled to multi-angle light scattering.** Size-exclusion chromatography coupled to multi-angle light scattering (SEC-MALS) experiments were performed using a Superdex 200 Increase 10/300 column (Cytiva), an ÄKTA fast protein liquid chromatography (FPLC) purifier (Cytiva) in line with a miniDAWN MALS detector (Wyatt Technology, Santa Barbara, CA. USA), and an Optilab digital refractive index detector (Wyatt Technology). All experiments were performed in 1X TBS. ASTRA VI software was used to combine these measurements and enable the absolute molar mass and extinction coefficient of the eluting GP, Fab, or GP–Fab complex to be determined[43,44].

**Composition gradient multi-angle light scattering.** Composition gradient multi-angle light scattering (CG-MALS) experiments were performed with a Calypso II composition gradient system (Wyatt Technology) to prepare different compositions of buffer, GP, and antibody and deliver to the miniDAWN detector and an online ultraviolet (UV) detector (Cytiva). The extinction coefficient obtained from the SEC-MALS experiment was used to measure the concentration of the GP during CG-MALS experiments. Polycarbonate filter membranes with 0.1 μM pore size (Millipore Sigma, Burlington, MA, USA) were installed in the Calypso system for sample and buffer filtration. GP was diluted to a stock concentration of 40–60 μg/mL in TBS. Fab was diluted to a stock concentration of 50–60 μg/mL in TBS. The automated Calypso method consisted of a dual-component "crossover" gradient to assess hetero-association between the GP and Fab. For each composition, a volume of 0.7 mL of protein solution was injected into the UV and MALS detectors until an equilibrium was reached

within the MALS flow cell and the flow stopped for 300–800 s. Data were collected, and analyses were performed with CALYPSO software. $GP_2$–3A6 association was measured in triplicate with two different preparations of GP and Fab.

## Protein assays

ELISA and a cell-based antibody-binding assay were performed with WT virus GPs or variants created via alanine scanning performed in accordance with standard protocols.

**Cell-based antibody binding assay.** To evaluate binding of mAbs to GP variants, HEK 293 T cells expressing full-length $GP_{1,2}$ or variants thereof were incubated with unlabeled mAbs at 10 μg/mL, followed by staining with DyLight 488 anti-human IgG and detection of fluorescence by microscopy. The binding of a control conformational mAb (ADI-15878)[9,45,46] was used as a control for GP expression levels. Secondary antibody binding alone was used as a negative control to assess background binding. In detail, HEK 293 T cells were plated at ≈$1 \times 10^5$ cells per well in 24-well plates treated with Poly-L-lysine solution (Millipore Sigma) 1 d prior to transfection. Cells were transiently transfected with 0.5 μg DNA per well using TransIT-LT1 transfection reagent (Mirus Bio, Madison, WI, USA). At 48 h post-transfection, cells were fixed with 4% paraformaldehyde (PFA; Electron Microscopy Sciences, Hatfield, PA, USA) in DPBS for 20 min. Cells were then incubated for 1 h at room temperature with 10 μg/mL primary mAbs in DPBS supplemented with 1% bovine serum albumin (BSA; Millipore Sigma). Cells were subsequently incubated at room temperature for 1 h with DyLight 488 anti-human IgG secondary antibody (SA5-10126; Thermo Fisher Scientific) and Hoechst 33342 (Invitrogen) in DPBS supplemented with 1% BSA. Finally, cells were imaged on a widefield fluorescence Axiovert 200 M Marianas microscope with a 10x/0.3 dry objective (ZEISS, Feasterville, PA, USA) or a confocal LSM780 microscope with a 10x/0.3 dry objective (ZEISS). Images were analyzed in QuPath[47]. Nuclei were segmented using the Hoechst image, and the objects were expanded by 5 μm to locate approximate cell boundaries. Anti-human IgG positive and -negative cells were counted using a trained object classifier. The classifier was optimized for the widefield and confocal images separately. Then, all data from three biological replicates were combined, and the 3A6-positive cell percentage was normalized against that obtained with ADI-15878.

**ELISA.** Microtiter plate wells were coated with purified recombinant WT or mutant $GP_{1,2\Delta TM/\Delta MLD}$ and incubated at room temperature for 1 h before blocking with 3% BSA (Millipore Sigma) in DPBS containing 0.05% TWEEN-20 (Thermo Fisher Scientific) for 1 h. Serial dilutions of mAb were applied to the wells and incubated for 1 h at room temperature. The bound antibodies were detected using Jackson Immuno Research Labs peroxidase-conjugated goat anti-human IgG (#109036006; Thermo Fisher Scientific) with horseradish peroxidase (diluted 1:4000) and 3,3',5,5'-tetramethylbenzidine (TMB) substrate (Thermo Fisher Scientific) before 50 μL of 1 N sulfuric acid solution (Thermo Fisher Scientific) was added to stop the reaction. Absorbance at 450 nm was then measured using a Spark microplate reader (Tecan, Männedorf, Switzerland). Half-maximal response ($EC_{50}$) values for mAb binding were determined using Prism 7 (GraphPad Software, Boston, MA, USA) after log-transformation of antibody concentrations using $EC_{50}$ shift nonlinear regression analysis.

**Plaque reduction assay using biologically contained EBOV.** A biologically contained EBOV, EbolaΔVP30 virus[21], was used to assess the impact of a P636S change on 3A6-mediated neutralization as previously described by Davis et al.[11]. Briefly, Ebola-GP-P636SΔVP30-eGFP virus was incubated with 10 μg/mL of mAb at 37 °C for 60 min. The virus/mAb mixture was then inoculated onto Vero VP30 cells, seeded the previous day in 12-well plates. After 60 min of incubation, cells

were washed to remove any unbound virus and overlaid with 1.25% methylcellulose media to allow for plaque formation. Seven days after infection, the number of plaques was quantified after immunochemistry staining with an antibody against the VP40 protein.

## Negative-stain electron microscopy

$GP_2$–3A6 complexes were obtained by incubating $GP_{1,2\Delta TM/\Delta MLD}$ with a three-fold molar excess of 3A6 overnight followed by purification using a Superdex 6 Increase 10/300 GL SEC column. The complexes were diluted to 0.01 mg/mL, and a volume of 4 μL of the complex solutions was applied to each freshly plasma-cleaned carbon-coated 400-mesh copper grid (Electron Microscopy Sciences) for 1 min. The solutions were blotted from the grids, followed by staining with 1% uranyl formate (Electron Microscopy Sciences) for 1 min. The stain was blotted from the grids, which were then air-dried before imaging. Images were collected on an FEI Titan Halo 300 kV electron microscope (Thermo Fisher Scientific) at a magnification of ×57,000 with a Falcon II camera. Contrast transfer function (CTF) correction, particle picking, 2-dimensional class averaging, and 3-dimensional reconstruction and refinement were all performed using cryoSPARC[48].

## Virion-like particle preparation, purification, and visualization

EBOV VLPs were prepared from HEK 293 T cells by co-expression of full-length $GP_{1,2}$ and VP40 essentially as previously described by Wan et al.[49], except that Gibco PBS (Thermo Fischer Scientific) was used instead of tris + NaCl + ethylenediaminetetraacetic acid (EDTA) (TNE) buffer. Clarification of supernatants from four 150 mm dishes was performed at 3000 × $g$ for 15 min at 4 °C. After density gradient purification, pellets were resuspended in 200 μL Gibco PBS.

## Cryogenic electron tomography

**Sample preparation, data collection, and tomogram reconstruction.** Fabs (1 mg/mL) were mixed with purified VLPs and 10 nm colloidal gold and incubated for 30–60 min at 4 °C. Different combinations of Fabs were prepared, imaged, and processed in parallel. The different mixtures (4 μL) were added to C-Flat 2/2 EM grids (Protochips, Morrisville, NC, USA) and vitrified by back-side blotting (4-s blotting time) using a LeicaGP cryo plunger (Leica, Deerfield, IL, USA) and stored in liquid nitrogen until imaging.

Cryogenic electron tomography data collection was performed essentially as previously described by Ke et al.[50] on a Titan Krios electron microscope equipped with Gatan Bioquantum energy filter and K3 direct electron camera (Thermo Fisher Scientific). The nominal magnification was 64,000×, giving a pixel size of 1.386 Å on the specimens. The defocus range was −2.0 to −4.5 μm, with a 0.25 μm step size (Supplementary Table 4).

**Subtomogram averaging.** Tomograms were reconstructed using IMOD[51], and the initial steps of subtomogram alignment and averaging were implemented using MATLAB (MathWorks) scripts and subTOM package (https://github.com/DustinMorado/subTOM/releases/tag/v1.1.0), which were derived from the TOM[52] and AV3[53] packages as previously described by Ke et al.[50].

To generate an initial starting model for each structure, 50–100 copies of GP were manually identified from VLP filaments that were down-scaled by 6× binning of the voxels and subjected to reference-free subtomogram alignment (Supplementary Fig. 5A). To identify the positions of all the particles on the surface of the VLP filaments, markers were placed manually along the filaments within the tube using the Volume Tracer function in UCSF Chimera (v.1.13.1)[54]. The radii of the filaments were determined centered at the membrane using the Pick Particle Chimera Plugin[55]. An oversampled cylindrical grid of points was generated on the filament surfaces with ≈8 nm of spacing, and subtomograms were extracted for all grid points with a box size of 64 pixels (≈50 nm) centered at a radius 10 nm above these grid

positions. Initial Euler angles were assigned to each subtomogram based on the orientation of the normal vectors relative to the cylinder surface. The oversampled positions were further aligned in the sub-TOM package. Duplicated particles and poorly aligned particles were removed from further analysis.

Subsequent processing was performed in RELION[56], as previously described by Ke et al.[50]. Our reported averages are the primary average structures for each sample. We used a tight mask to obtain a higher resolution of the GP structure, while a cylindrical mask including the membrane-proximal region to obtain a better overview map including the membrane region where the Fab fragment is located (Supplementary Fig. 5B). The VLP $GP_{1,2}$–3A6 structure was refined to 17.7 Å with 9602 particles from 26 tomograms; the VLP $GP_{1,2}$–KZ52 structure was refined to 8.9 Å (tight mask) and 12.7 Å (cylinder mask) with 13,520 particles from 18 tomograms; and the VLP $GP_{1,2}$–KZ52/3A6 structure was refined to 7.4 Å (tight mask) and 9.3 Å (cylinder mask) with 40,072 particles from 42 tomograms (Supplementary Fig. 5C, Supplementary Table 4). Contours shown in Figs. 3 and 4 were normalized by eye so that the $GP_{1,2}$ densities are as equivalent as possible between different maps.

## AlphaFold 2 modeling
We used ColabFold[25] to generate complete structures of the EBOV GP using the AlphaFold 2 multimer protocol[24]. Four independent analyses were performed to generate a total of 20 models. Two experiments used previously solved structures of EBOV GP (PDB 6VKM or 5JQ3) as a template for modeling and two experiments used the PDB100 as templates for modeling.

## Alanine scanning and antibody binding test
Alanine scanning was performed by introducing single alanine residues (alanines were changed to serines) into $GP_{1,2\Delta MLD}$ region 627–639 via site-directed mutagenesis of $GP_{1,2\Delta MLD}$-encoding plasmid. The plasmid clones were individually arrayed into 384-well plates and transfected into HEK 293T cells. Protein variants were cell-surface-expressed for 22 h[20]. The indicated mAbs were incubated with the cells for 1 h before an Alexa Fluor 488-conjugated secondary antibody (Thermo Fisher Scientific) was added. Antibody binding was assessed by detection of cellular fluorescence with a high-throughput flow cytometer (Intellicyt, Albuquerque, NM, USA). Background fluorescence was measured in vector-transfected control cells and mAb reactivity against the variants was calculated with respect to reactivity with $GP_{1,2\Delta MLD}$ by subtracting the signal from mock-transfected controls and normalized to signals from WT $GP_{1,2\Delta MLD}$-transfected controls. Residues predicted to be involved in the epitope were identified when mAb and variant did not react. But, when reactivity of other control mAbs was observed, GP variants that were misfolded or were expressed at low levels were excluded.

## Animal studies
Animal exposure and treatment experiments using infectious EBOV were performed in the biosafety level 4 (BSL-4) laboratory at the Integrated Research Facility at Fort Detrick (IRF-Frederick), Division of Clinical Research (DCR), National Institutes for Allergy and Infectious Diseases (NIAID), National Institutes of Health (NIH) under accreditation (000777) by the Association for Assessment and Accreditation of Laboratory Animal Care (AAALAC), Laboratory Animal Welfare approval (D16-00602) by the Public Health Service (PHS), and United States Department of Agriculture (USDA) registration (51-F-0016). Animal experiments were approved by the NIAID DCR Animal Care and Use Committee (ACUC) and followed the recommendations provided in the Guide for the Care and Use of Laboratory Animals.

**Domesticated guinea pigs.** Hartley strain domesticated guinea pigs (*Cavia porcellus* (Linnaeus, 1758)) of both sexes, ages 6–8 weeks (Charles

River Laboratories, Wilmington, MA, USA), were assigned to five groups of six animals (three males and three females) each. All animals were exposed IP to 1000 PFU of domesticated guinea pig-adapted Ebola virus/UTMB/C.porcellus-lab/COD/1976/Yambuku-Mayinga-GPA on Day 0. Animals in each group were injected IP on Day 3 with 5 mg of 3A6, 1A2, 7G7, or 42-2D2 anti-FLUAV IgG in 3 mL of DPBS or administered no treatment. Animals were observed daily for clinical signs of disease and were assigned a clinical score of 0–3 (0=none; 1=mild; 2=moderate; 3=severe). Animals reaching endpoint criteria (score of 3) were humanely euthanized. Weight was recorded daily starting 1 d before exposure until all animals recovered from disease (Day 15), then twice weekly until the study endpoint on Day 28. Blood was collected twice, on Day 6 after exposure and at the time of euthanasia.

**Rhesus monkeys.** Four rhesus monkeys (*Macaca mulatta* (Zimmermann, 1780)) of both sexes, aged 4–5 years (WorldWide Primates, Miami, FL, USA) were single-housed and acclimated to BSL-4 conditions prior to virus exposure. On Day 0, monkeys were sedated using IM injection of 15 mg/kg of Ketamine HCl (KetaThesia; Henry Schein, Melville, NY, USA) and exposed IM with a target dose of 1000 PFU of Ebola virus/H.sapiens-tc/COD/1995/Kikwit-9510621 (NR-50306, Lot 9510621, ≥95% 7U abundance at the *GP* editing site; BEI Resources, Manassas, VA, USA); the same dose and lot of this virus previously resulted in fatal outcome at 5–8 d post-exposure in 12 out of 12 untreated rhesus monkeys ("historical controls"[57]). On Day 4 and Day 7 after exposure, monkeys 1–3 received 25 mg/kg of 3A6 in DPBS (provided by Chakravarthy Reddyvia) by intravenous infusion, and the control monkey received an equivalent volume of DPBS. All monkeys were observed for the development of clinical signs of EBOV infection and scored daily according to a four-point scoring scale. Physical examination and blood collection were conducted on the monkeys twice prior to exposure (Day -14 and Day -7) and at 4, 7, 9, 12, 21, and 28 d after exposure. Complete blood counts with reticulocytes and differential was analyzed on a Sysmex XT-2000iV hematology instrument (Sysmex America, New York, NY, USA). Sera were obtained after separation at room temperature and centrifugation for 15 min at 1500 × *g* followed by analysis using the Piccolo general chemistry 13 panel on a Piccolo Xpress analyzer (Abaxis, NJ, USA). Prothrombin and activated partial thromboplastin times were measured on a CS-2500 system automated coagulation analyzer (Sysmex America). Infectious titers were determined in sera using an Avicel-based crystal violet stain plaque assay on Vero E6 cell culture monolayers (BEI Resources) as previously described by Shurtleff et al, 2012[58]. Sera were inactivated in TRIzol LS (Thermo Fisher Scientific) according to the manufacturer's instructions and nucleic acid extracted using the QIAamp Viral RNA Mini Kit (Qiagen, Germantown, MD, USA). The Critical Reagents Program (CRP) EZ1 RT-PCR kit assay (BEI Resources) was used in accordance with manufacturer's instructions[33] on an Applied Biosystems 7500 FastDx Real-Time PCR instrument (Thermo Fisher Scientific) to quantify EBOV nucleic acids in sera and to transform results into $\log_{10}$ genome equivalents (GE) per mL of sample. The control monkey succumbed on Day 8 whereas the treated monkeys underwent elective euthanasia approximately 3 mo after virus exposure.

## Statistical analysis
Statistical details of experiments, including numbers of replicates and measures of precision (standard deviation, SD), can be found in the figure legends, figures, results, and methods. Dose-response ELISA curves were fit to a $EC_{50}$ shift by nonlinear regression analysis. All analyses were performed with Prism 7.

## Resource availability
Further information and requests for resources and reagents should be directed to and will be fulfilled by the lead contact, Erica Ollmann Saphire (erica@lji.org).

**Reporting summary**

Further information on research design is available in the Nature Portfolio Reporting Summary linked to this article.

## Data availability

Structure factors and associated model coordinates generated in this study have been deposited in the Protein Data Bank (PDB; http://www.rcsb.org) under PDB accession codes 7RPU (3A6–stalk–MPER, https://doi.org/10.2210/pdb7RPU/pdb) and 7RPT (apo3A6, https://doi.org/10.2210/pdb7RPt/pdb). Tomographic reconstructions have been deposited in the Electron Microscopy Data Bank (EMDB; https://www.ebi.ac.uk/emdb/) under accession codes EMD-45833, EMD-45834, EMD-45835, EMD-45836 https://www.ebi.ac.uk/pdbe/entry/emdb/EMD-45836), and EMD-45837.

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

## Acknowledgements

We are grateful to Dustin Morado, Kun Qu, and Joaquin Oton (Medical Research Council Laboratory of Molecular Biology) for support in collecting and/or processing cryogenic electron tomography data and to Stanford Synchrotron Light Source (SSRL) and Advanced Photon Source (APS) for assistance in collection of X-ray data. We thank Dawid Zyla (La Jolla Institute for Immunology) for assistance in the generation of AlphaFold models. We thank Anya Crane (National Institutes of Health [NIH], National Institute of Allergy and Infectious Diseases [NIAID], Integrated Research Facility at Fort Detrick, Frederick, MD, USA [IRF-Frederick]) for critically editing the manuscript and Jiro Wada (NIH NIAID IRF-Frederick) for creating figures. We gratefully acknowledge Peter B. Jahrling and Lisa E. Hensley (NIH NIAID IRF-Frederick) for supporting the studies involving animals. The work for this study was supported by NIH NIAID U19 AI142790 (E.O.S. and C.W.D.), DARPA contract W31P4Q-14-1-0010 (C.W.D. and R.A.), NIH NIAID contract HHSN272201400058C to B.J.D., and UK Medical Research Council (MC_UP_1201/16), the European Research Council (ERC) under the European Union's Horizon 2020 research and innovation program (ERC-CoG-648432 MEMBRANEFUSION), and the Max Planck Society (J.B.). This research was further supported through the NIH/NIAID prime contract with Battelle Memorial Institute (Contract No. HHSN272200700016I) and subsequently with Laulima Government Solutions (Contract No. HHSN272201800013C). G.W. and M.R.H. performed this work as employees under these contracts. J.H.K. performed this work partly as an employee under the Battelle Memorial Institute contract and partly as an employee of Tunnell Government Services, formerly a subcontractor of Battelle Memorial Institute and now of Laulima Government Solutions. This work has been further funded in whole or in part with federal funds from the National Cancer Institute, National Institutes of Health, under Contract No. HHSN261201500003I, Task Order No. HHSN26100043 and Contract No. 75N91019D00024, Task Order No. 75N91019F00130 (I.C.). S.M. was funded by an Imaging Scientist grant (2019-198153) from the Chan Zuckerberg Initiative. Use of the SSRL SLAC National Accelerator Laboratory is supported by the US Department of Energy (DOE) Office of Science Basic Energy Sciences (BES) program under Contract No. DE-AC02-76SF00515. The SSRL Structural Molecular Biology Program is supported by the DOE Biological and Environmental Research (BER) program and by the NIH National Institute of General Medical Sciences (NIGMS; P41GM103393). This research also used resources of the Advanced Photon Source (APS), a DOE Office of Science User Facility operated for the DOE Office of Science by Argonne National Laboratory (ANL) under Contract No. DE-AC02-06CH11357. The General Medical Sciences and Cancer Institute Structural Biology Facility at the Advanced Photon Source (GM/CA) project has been funded in whole or in part with Federal funds from the National Cancer Institute (ACB-12002) and the NIGMS (AGM-12006). The Eiger 16M detector at the ANL X-ray Science Division (XSD) was funded by NIH grant S10 OD012289. Opinions, interpretations, conclusions, and recommendations are those of the authors and are not necessarily endorsed by the U.S. Army. The funding sources were not involved in study design; in the collection, analysis, and interpretation of data; in the writing of the report; and in the decision to submit the paper for publication. The views and conclusions contained in this document are those of the authors and should not be interpreted as necessarily representing the official policies, either expressed or implied, of the U.S. Department of Health and Human Services or of the institutions and companies affiliated with the authors, nor does mention of trade names, commercial products, or organizations imply endorsement by the U.S. Government.

## Author contributions

Conceptualization, K.M.H., Z.L.S., J.A.G.B., L.E.D., G.W., M.H.H., C.W.D., R.A., and E.O.S. Methodology, K.M.H., Z.L.S., Z.K., E.D., B.D., and E.O.S. Investigation, K.M.H., Z.L.S., Z.K., P.J.H., S.A. E.D., L.E.D., A.G., C.C., C.H., S.M., M.J.N., and X.X. Formal analysis, K.M.H., Z.L.S., X.Y., Z.K., J.A.G.B., L.E.D., G.W., E.O.S., Writing—Original Draft, Z.L.S. and E.O.S. Writing—Review & Editing, K.M.H., Z.L.S., Z.K., S.L.S., X.Y., I.C., J.H.K., J.A.G.B., G.W., R.A., and E.O.S. Visualization, K.M.H., Z.L.S, Z.K., X.Y., S.M. Supervision, B.D., Y.K., J.A.G.B., and E.O.S. Resources, B.D., M.H., L.M.B., I.C., and J.H.K. Funding Acquisition, J.A.G.B., C.W.D., R.A., and E.O.S.

## Funding

## Competing interests

A.G., E.D., and B.J.D. are employees of Integral Molecular, and B.J.D. is a shareholder in that company. The remaining authors declare no competing interests.

## Ethics approval

Animal exposure and treatment experiments using infectious EBOV were performed in registered an ethics accredited biosafety level 4 (BSL-4) laboratories with appropriate approvals.

## Additional information

[1]Center for Vaccine Innovation, La Jolla Institute for Immunology, La Jolla, CA, USA. [2]Division of Structural Studies, Medical Research Council Laboratory of Molecular Biology, Cambridge, UK. [3]Department of Cell and Virus Structure, Max Planck Institute of Biochemistry, Martinsried, Munich, Germany. [4]Influenza Research Institute, Department of Pathobiological Sciences, School of Veterinary Medicine, University of Wisconsin, Madison, WI, USA. [5]Integrated Research Facility at Fort Detrick, National Institute of Allergy and Infectious Diseases, National Institutes of Health, Fort Detrick, Frederick, MD, USA. [6]Microscopy Core, La Jolla Institute for Immunology, La Jolla, La Jolla, CA, USA. [7]Integral Molecular, Philadelphia, PA, USA. [8]Department of Microbiology and Immunology, Emory Vaccine Center, Atlanta, GA, USA. [9]Zalgen Labs LLC, Frederick, MD, USA. [10]Clinical Monitoring Research Program Directorate, Frederick National Laboratory for Cancer Research, Frederick, MD, USA. [11]Division of Virology, Institute of Medical Science, University of Tokyo, Tokyo, Japan. [12]The Research Center for Global Viral Diseases, National Center for Global Health and Medicine Research Institute, Tokyo, Japan. [13]Pandemic Preparedness, Infection and Advanced Research Center (UTOPIA), University of Tokyo, Tokyo, Japan. [14]Department of Medicine, University of California San Diego, La Jolla, CA, USA. [15]Department of Molecular Biosciences, the University of Texas at Austin, Austin, TX, USA. [16]Guangzhou Regenerative Medicine and Health—Guangdong Laboratory, Guangzhou Institutes of Biomedicine and Health, Chinese Academy of Sciences, Science Park, Guangzhou, Guangdong Province, China. [17]Arcturus Therapeutics, San Diego, CA, USA. [18]Present address: Eli Lilly, San Diego, CA, USA. [19]Present address: Department of Biochemistry, University of Toronto, Toronto, ON, Canada. [20]Present address: Vall d'Hebron Institute of Oncology, Hospital del Mar Research Institute, Barcelona, Spain. [21]These authors contributed equally: Zhe Li Salie, Zunlong Ke. ✉e-mail: kuhnjens@niaid.nih.gov; briggs@biochem.mpg.de; gabriella.worwa@nih.gov; cwdavi2@emory.edus; rahmed@emory.edu; erica@lji.org

