## [Transparent Peer Review file · Nature Communications]

Anti-Ebola virus mAb 3A6 protects highly viremic animals from fatal outcome via binding GP(1,2) in a position elevated from the virion membrane

Corresponding Author: Professor Erica Saphire

Version 0:

Reviewer comments:

Reviewer #1

(Remarks to the Author)

In the manuscript “Anti-Ebola virus mAb 3A6 with unprecedented potency protects highly viremic animals from fatal outcome and physically lifts its glycoprotein target from the virion membrane”, Salie, et al describe new studies on a previously isolated and characterized mAb 3A6 that targets the membrane proximal external region (MPER) of Ebola virus GP. They present a structural analysis of 3A6 by X-ray crystallography and cryogenic electron tomography (cryo-ET) and an assessment of therapeutic efficacy when administered as monotherapy post-infection in guinea pigs and macaques. 3A6 is found to be highly potent in that half the dose of 3A6 provides a similar level of protection as monotherapy with standard-of-care ansuvimab (mAb114) monotherapy that targets the receptor binding region on GP1, barring differences in days administered. As such, 3A6 has important potential for countermeasure development.

In the structural studies, they report crystal structures of 3A6 Fab both unbound and in complex with an EBOV GP MPER peptide revealing 3A6 recognizes a helical epitope of the stem/MPER that is partially unraveled at its C-terminus. For the cryo-ET studies, they assess 3A6 binding to a full-length GP1,2 on the surface of EBOV virus like particles (VLPs) and suggest that the antibody “lifts” GP from the membrane surface. While significant for addressing 3A6 binding to GP in situ, namely in the context of the VLP membrane, the somewhat varied resolutions, the lack of an unbound GP1,2 control structure, and the comparison to only one other antibody-bound structure, are of concern given the conclusions that are drawn.

Comments:

1. Assessment of “lift” from the viral membrane by 3A6 is based on comparison against KZ52 bound GP structure but not against an unbound GP control or against any other Fab-bound GPs. As such, there is a built-in assumption that KZ52 bound GP represents a baseline state of GP in this context. However, the proper baseline for assessment of such displacement should be an unbound GP structure.
2. It is not clear if the subtomogram data yielded any other GP classes or only the ones presented?
3. Relatedly, type I viral membrane fusion proteins often exhibit multiple tilt angles relative to the viral membrane. The reconstructions presented appear to be strictly at ~90 deg. Were there other GP tilt angles present? Could such tilt angles influence assessments of “lift” off the viral membrane?
4. Were there only GPs with three bound Fabs or were populations with 1 or 2 bound Fabs also observed?
5. cryo-ET maps were apparently not deposited in EMDB?
6. Comparison of cryo-ET maps are a central aspect of the structural analysis. Since the structures were at different resolutions, how were map contours normalized to be comparable across the three structures?
7. BDBV223 is another MPER-specific antibody that targets an overlapping epitope on MPER. No pairwise structural comparisons are performed between 3A6 and BDBV223 crystal structures, which is a missed opportunity. Do they interact

with the same sets of residues on GP MPER? How do they differ in their recognition of the epitope?

8. Figure 3: docking of 3A6-MPER complex onto 5JQ7 not appropriate since 5JQ7 has a foldon motif at its C-terminus which likely constrains the stalk as a whole, as noted by the authors themselves.

9. The impression is given that the epitope is occluded by the viral membrane (e.g., see lines 288, 290). While possible, what direct evidence is there that the epitope is in fact occluded by the viral membrane (other than proximity)?

10. Line 223: Statement that this is "...a first-in-class antibody that appears to perform physical work" poses the question of how the term "physical work" is defined in this context? Could the possible displacement of the head region of GP from membrane be a result of unraveling of helical portions of the stem (or related bundles) either within or outside the 3A6 epitope?

11. Relatedly, use of the term "lift" implies the antibody is extracting GP from the viral membrane, but that is not fully substantiated.

12. Line 304: Statement that 3A6 is founding member of this immunotherapeutic class is confusing given previous statements in the manuscript regarding the BDBV223 antibody.

13. Supplementary Figure 2 title states that residues D632 and P636 of the Ebola virus glycoprotein MPER are key for mAb 3A6 binding, but these positions do not appear to be addressed in the figure.

14. Supplementary Table 2 does not state what experiment the data reflect.

Reviewer #2

(Remarks to the Author)

Salie and colleagues describe the neutralization mechanism of ab 3A6 that recognizes the stalk and MPER of Ebola virus GP. The authors propose that 3A6 lifts GP from the membrane. Furthermore, data is presented that indicates that 3A6 is the most potent mAb yet used in monotherapy that has a beneficial effect at high viremia and advanced disease stages.

The major findings are:

The crystal structure of 2A6 in complex with the stalk-MPER epitope adopting a partly helical conformation for residues 627 to 639.

Residues critical for 2A6 binding were identified. Important residues D632 and P636 are conserved in EBOV and SUDV. Modeling of the MPER epitope conformation onto native GP structures indicates clashes with neighboring protomers upon 3A6 binding. However, in vitro studies show that GP1-2deltaTM binds 3 3A6 Fabs. Tomography of membrane-anchored GP confirmed binding of 3 Fabs, which extended the overall height of GP by 3 nm as compared to GP-KZ52 Fab binding.

Post exposure 3A6 Ab treatment of EBOV infected guinea pigs and macaques reversed EBOV infection.

In summary, the manuscript describes the structure of an EBOV MPER-specific antibody, proposes a mechanism of action and its therapeutic application. The work will be of broad interest to the virology community and will help to extend therapeutic possibilities against lethal filovirus infections.

The following points should be addressed prior to publication in Nature Communications:

The binding model in Fig 3 suggests a large membrane-Ab interface. Does 3A6 bind to membranes?

The number of animals/macaques is quite low. I suggest to tone down the effect of the Ab treatment on clinical parameters.

Previous studies testing the effect of post exposure treatment used treatment at Day 5 post infection. Is there any specific reason why day 4 was chosen. Is disease progression significant from day 4 to 5? The control monkey died at day 8.

Minor points:

Line 93: Citation 8 describes the structure of MPER-TM of GP2, but does not establish GP2 as a class I fusion molecule.

Line 161: D632A and P636A mutations produced a <20 % reduction in 3A6 Fab binding?? Fig 3C shows a less than 20% binding.

Line 222: ... and identify 3A6 IgG as a first-in-class antibody that appears to perform physical work. This statement is not correct, because the same mechanism has been proposed for HIV-1 MPER antibody 4E10. MPER is likely a dynamic structure and an extended MPER may be locked by 3A6. Do all GPs show the same height on the VLPs in the absence of 3A6?

Line 292: The manuscript contains no data that allows to claim that a portion of the 3A6 epitope is indeed embedded in the membrane.

Supplementary Fig 4 has 5 panels A, B, C, D, E but the main text refers only to Supplementary Fig 4.

Supplementary Fig 5 has two panels A and B, but the main text refers only to B.

Version 1:

Reviewer comments:

Reviewer #1

(Remarks to the Author)

In the revised manuscript the authors have adequately addressed all previous comments.

Reviewer #2

(Remarks to the Author)

The concerns have been reasonably addressed in the revisions and I support publication in its current form.

REVIEWER COMMENTS

Reviewer 1:

In the manuscript “Anti-Ebola virus mAb 3A6 with unprecedented potency protects highly viremic animals from fatal outcome and physically lifts its glycoprotein target from the virion membrane”, Salie, et al describe new studies on a previously isolated and characterized mAb 3A6 that targets the membrane proximal external region (MPER) of Ebola virus GP. They present a structural analysis of 3A6 by X-ray crystallography and cryogenic electron tomography (cryo-ET) and an assessment of therapeutic efficacy when administered as monotherapy post-infection in guinea pigs and macaques. 3A6 is found to be highly potent in that half the dose of 3A6 provides a similar level of protection as monotherapy with standard-of-care ansuvimab (mAb114) monotherapy that targets the receptor binding region on GP1, barring differences in days administered. As such, 3A6 has important potential for countermeasure development.

In the structural studies, they report crystal structures of 3A6 Fab both unbound and in complex with an EBOV GP MPER peptide revealing 3A6 recognizes a helical epitope of the stem/MPER that is partially unraveled at its C-terminus. For the cryo-ET studies, they assess 3A6 binding to a full-length GP_{1,2} on the surface of EBOV virus like particles (VLPs) and suggest that the antibody “lifts” GP from the membrane surface. While significant for addressing 3A6 binding to GP in situ, namely in the context of the VLP membrane, the somewhat varied resolutions, the lack of an unbound GP_{1,2} control structure, and the comparison to only one other antibody-bound structure, are of concern given the conclusions that are drawn.

Comments:

1. Assessment of “lift” from the viral membrane by 3A6 is based on comparison against KZ52 bound GP structure but not against an unbound GP control or against any other Fab-bound GPs. As such, there is a built-in assumption that KZ52 bound GP represents a baseline state of GP in this context. However, the proper baseline for assessment of such displacement should be an unbound GP structure.

Our response: We appreciate the reviewer’s concern. We acknowledge that an unbound GP appears to be an obvious control, but unbound GP could not be resolved to a reasonable resolution from our datasets. As we state in the paper,

Line 233: “...to represent the state of GP_{1,2} in the absence of 3A6, we complexed VLPs with the Fab of the well-characterized mAb KZ52, which binds to the fusion peptide of EBOV GP_{1,2}”

Importantly, we do not only compare 3A6 with KZ52 bound to the glycoprotein, but we also show a glycoprotein bound simultaneously to both 3A6 and KZ52. Where both antibodies are bound, the position matches that of 3A6 alone. We state,

Line 242: “The KZ52 epitope at the base of GP₁ is sufficiently separated from the stalk–MPER 3A6 binding site and its binding does not disrupt the native structure of GP_{1,2}.” We also state, (line 248) “We conclude that 3A6 binding induces this vertical displacement since VLP GP_{1,2} complexed with both 3A6 and KZ52 Fabs together resulted in a similarly lifted GP_{1,2} (Fig. 4C) with densities that overlapped with VLP GP_{1,2}–3A6 (Fig. 4D).”

Based on this, in our opinion, the only reasonable interpretation of the three structures we present is that 3A6 binding causes the glycoprotein to be positioned farther from the membrane.

2. It is not clear if the subtomogram data yielded any other GP classes or only the ones presented?
Our response: We did not perform deep classification of the subtomogram averaging data, which can be challenging in particular for membrane-proximal structures. Instead, we present the primary bulk class, excluding only poorly aligned particles. These structures therefore represent the primary structure for each sample. We have added a sentence to the methods for clarification:

Line 242: “The strong Fab density in each reconstruction suggests that the bulk population of GP_{1,2} is fully occupied by the bound Fab(s), however we have not performed a classification aimed at resolving minority populations with fewer numbers of Fabs.”

3. Relatedly, type I viral membrane fusion proteins often exhibit multiple tilt angles relative to the viral membrane. The reconstructions presented appear to be strictly at ~90 deg. Were there other GP tilt angles present? Could such tilt angles influence assessments of “lift” off the viral membrane?

Our response: We did not separate particles according to tilt, but as we compared the position of the glycoprotein with the position of the membrane directly at the base of the glycoprotein, perpendicular to the glycoprotein symmetry axis, we do not believe that variable tilt angles would influence our assessment of lift.

Further, the reconstructions presented reflect only those classes with enough particles for reconstructing a 3D volume. We agree that viral glycoproteins are highly dynamic, with the ability to tilt about the surface of the membrane. This dynamic nature likely contributes to our inability to capture a resolvable image of cell-surface displayed glycoprotein in the absence of antibody. In fact, the inherent ability of the glycoprotein to tilt may be a requirement for exposure of the 3A6 epitope. However, once bound by mAb, the glycoprotein may become fixed in the 90° position. A similar phenomenon may also be present for KZ52 whose binding restricts the radial movement of the glycoprotein on the surface. Mechanistic studies of how this protein behaves in the presence of mAbs that target different epitopes are of great interest and something we would like to further investigate.

4. Were there only GPs with three bound Fabs or were populations with 1 or 2 bound Fabs also observed?
Our response: We have clarified this in the text:

Line 242: “The strong Fab density in each reconstruction suggests that the bulk population of GP_{1,2} is fully occupied by the bound Fab(s), however we have not performed a classification aimed at resolving minority populations with fewer numbers of Fabs.”

5. cryo-ET maps were apparently not deposited in EMDB?

Our response: We apologize for this oversight. The maps have been deposited in EMDB and accession codes have been included in the revised manuscript (line 666)

6. Comparison of cryo-ET maps are a central aspect of the structural analysis. Since the structures were at different resolutions, how were map contours normalized to be comparable across the three structures?

Our response: We thank the reviewer for this excellent question and we apologize for not clearly explaining this in the methods. We have updated the methods section to include additional details: Line 576: “Contours were normalized by eye so that the GP_{1,2} densities are as equivalent as possible between

different maps.” Importantly, conclusions about lift would not be altered by small changes in the contour level.

7. BDBV223 is another MPER-specific antibody that targets an overlapping epitope on MPER. No pairwise structural comparisons are performed between 3A6 and BDBV223 crystal structures, which is a missed opportunity. Do they interact with the same sets of residues on GP MPER? How do they differ in their recognition of the epitope?

Our response: We thank the reviewer for this suggestion. We agree that it is a valuable comparison to make and have now included an additional supplementary figure detailing the binding sites and modes for these mAbs and have updated the text accordingly.

8. Figure 3: docking of 3A6-MPER complex onto 5JQ7 not appropriate since 5JQ7 has a foldon motif at its C-terminus which likely constrains the stalk as a whole, as noted by the authors themselves.

Our response: We agree that use of this structure for comparative purposes is not entirely useful in the context of native, full-length glycoprotein. Here, we use it to illustrate that three copies of 3A6 could not bind to a three-helix bundle of the stalk-MPER. Instead, these helices must separate for full-occupancy binding. It is not yet known whether the stalk-MPER exists as a three-helix bundle in native unbound glycoprotein (as found for the related Marburg virus glycoprotein) or if it is a more open arrangement. We have updated the text to better reflect the conclusions that can be drawn from this comparison:

Line 205: “It is not known whether inclusion of exogenous trimerization domains induce a non-native association of the HR2/MPER helices or if the stalk region adopts a more open and flexible conformation in the context of membrane-bound GP_{1,2}. Certainly, superimposition of the 3A6-stalk-MPER structure onto a fibrin-stabilized GP_{1,2ΔTM/ΔMLD} trimer (PDB 5JQ7) revealed steric clashes of the bound 3A6 with the other two GP₂ monomers of stalk-MPER (Supplementary Fig. 4A). Hence, the tightly bundled conformation of GP₂ HR2 observed in this crystal structure is incompatible with Fab binding”

9. The impression is given that the epitope is occluded by the viral membrane (e.g., see lines 288, 290). While possible, what direct evidence is there that the epitope is in fact occluded by the viral membrane (other than proximity)?

Our response: We thank the reviewer for this question. We observed that the angle at which 3A6 approaches the stalk of the glycoprotein would be incompatible with previous models of transmembrane glycoprotein unless the glycoprotein were lifted away from the membrane, but we do not have a direct measurement of whether the epitope residues are occluded and, if so, for how long or how often. Perhaps previous models underestimated the height, or perhaps glycoprotein samples compact and extend in a “breathing” motion, like those observed for HIV-1 and influenza A virus. Our original Fig. 3 displayed a model of the glycoprotein residues up to and including those visualized in our stalk-MPER peptide-bound 3A6 structure (residue 639). However, the MPER is predicted to extend to residue 651. The structure of these ≈15 residues is not yet defined in the context of the full-length GP. To better understand how 3A6 may access its epitope, we used AlphaFold 2 to model the glycoprotein, including MPER, interfacial region (IFR), and transmembrane domain (new Supplementary Fig. 6 and Fig. 4). Notably, our modeling suggests the glycoprotein may sample both compacted and extended conformations of the MPER-IFR. The solution structure of the MPER transmembrane domain of the glycoprotein aligns well with the compact form, and the compact model of the full-length glycoprotein can be successfully docked to the KZ52-bound glycoprotein map but shows poor fit to the 3A6-bound glycoprotein map. Conversely, we find that the extended model of the glycoprotein fits well into the 3A6-bound map.

10. Line 223: Statement that this is “...a first-in-class antibody that appears to perform physical work” poses the question of how the term “physical work” is defined in this context? Could the possible displacement of the head region of GP from membrane be a result of unraveling of helical portions of the stem (or related bundles) either within or outside the 3A6 epitope?

Our response: We agree with the reviewer that our original statement of “physical work” was not well-defined. We have modified the text to clarify how we believe 3A6 may access this membrane-proximal epitope and stabilize this region of GP in an extended upright state and have deleted the reference to physical work or first-in-class.

11. Relatedly, use of the term “lift” implies the antibody is extracting GP from the viral membrane, but that is not fully substantiated.

Our response: We thank the reviewer for this constructive feedback. We agree that use of the word “lift” may be a misleading interpretation of our data. We have revised the manuscript to reflect that 3A6 stabilizes the glycoprotein in a lifted conformation. We do not have experimental evidence that 3A6 binding requires extraction of the epitope from the membrane. Indeed, the absence of 3A6 binding to untransfected cells suggests that the antibody is unlikely to bind to membranes in the course of epitope engagement. As stated above, we have modeled a potential path for epitope recognition by 3A6.

12. Line 304: Statement that 3A6 is founding member of this immunotherapeutic class is confusing given previous statements in the manuscript regarding the BDBV223 antibody.

Our response: Thank you for pointing out these contradictory statements. We have revised the text accordingly and removed references to “first-in-class”.

13. Supplementary Figure 2 title states that residues D632 and P636 of the Ebola virus glycoprotein MPER are key for mAb 3A6 binding, but these positions do not appear to be addressed in the figure.

Our response: Thank you for bringing this error to our attention. This figure call-out was incorrect and should only include reference to Supplementary Table 2. Data for these changes are shown in Fig. 2. In addition, we modified Figure 1A to show D632.

14. Supplementary Table 2 does not state what experiment the data reflect.

Our response: Thank you for bringing this to our attention. We have modified the title for Supplementary Table 2 (now titled “Flow cytometric analysis of 3A6 binding to EBOV GP_{1,2ΔMLD} bearing the indicated alanine variant”).

Reviewer 2:

Salie and colleagues describe the neutralization mechanism of ab 3A6 that recognizes the stalk and MPER of Ebola virus GP. The authors propose that 3A6 lifts GP from the membrane. Furthermore, data is presented that indicates that 3A6 is the most potent mAb yet used in monotherapy that has a beneficial effect at high viremia and advanced disease stages.

The major findings are:

(1) The crystal structure of 2A6 in complex with the stalk-MPER epitope adopting a partly helical

conformation for residues 627 to 639. (2) Residues critical for 2A6 binding were identified. Important residues D632 and P636 are conserved in EBOV and SUDV. (3) Modeling of the MPER epitope conformation onto native GP structures indicates clashes with neighboring protomers upon 3A6 binding. However, in vitro studies show that GP1-2deltaTM binds 3 3A6 Fabs. Tomography of membrane-anchored GP confirmed binding of 3 Fabs, which extended the overall height of GP by 3 nm as compared to GP-KZ52 Fab binding. (4) Post exposure 3A6 Ab treatment of EBOV infected guinea pigs and macaques reversed EBOV infection.

In summary, the manuscript describes the structure of an EBOV MPER-specific antibody, proposes a mechanism of action and its therapeutic application. The work will be of broad interest to the virology community and will help to extend therapeutic possibilities against lethal filovirus infections. The following points should be addressed prior to publication in Nature Communications:

1. The binding model in Fig 3 suggests a large membrane-Ab interface. Does 3A6 bind to membranes?

Our response: We thank the reviewer for this question; we agree that our initial interpretations of our structural data did not provide the direct evidence needed for us to make such a conclusive statement. We observed that the angle at which 3A6 approaches the stalk of the glycoprotein would be incompatible unless the GP were lifted away from the membrane. Indeed, we do not observe 3A6 binding to untransfected cells (Supplementary Fig. 2), suggesting that the mAb does not interact alone with cellular membranes in isolation.

2. The number of animals/macaques is quite low. I suggest to tone down the effect of the Ab treatment on clinical parameters. Previous studies testing the effect of post exposure treatment used treatment at Day 5 post infection. Is there any specific reason why day 4 was chosen. Is disease progression significant from day 4 to 5? The control monkey died at day 8.

Our response: We thank the reviewer for this insightful comment and suggestion. In our experience, the 1,000-PFU exposure model accurately recapitulates a fulminant disease course in which there are significant differences in the day-to-day (even hour-to-hour) disease progression. Others' studies have also reported that some control NHPs succumb on Day 6. Hence, initiating treatment Day 5 could mean treating already highly viremic and sick animals. Since 3A6 had (prior to our study) not been evaluated in the NHP model of infection, we chose a more conservative treatment regimen by using two doses and starting treatment on Day 4 rather than Day 5. The study presented in our manuscript was intended as a proof of concept for the utility of 3A6 treatment of NHPs in the post-exposure setting. We have included these caveats in the revised manuscript. We have also revised the discussion to acknowledge that follow-up studies should include a larger cohort of NHPs with variable dosing:

Line 311: "These preliminary results demonstrate that post-exposure dosing of 3A6 IgG alone reverses the course of EBOV infection and protects against fatal outcomes. Additional studies, including a larger cohort of NHPs and differential dosing (i.e., lower dose of antibody, variation in the number of doses and day of treatment) will further illuminate the therapeutic impact of 3A6 treatment."

Minor points:

Line 93: Citation 8 describes the structure of MPER-TM of GP2, but does not establish GP2 as a class I fusion molecule.

Our response: Thank you for bringing this error to our attention. We have updated the text with the appropriate reference.

Line 171: D632A and P636A mutations produced a <20 % reduction in 3A6 Fab binding?? Fig 3C shows a less than 20% binding.

Our response: Thank you for your attention to detail. We have amended the text to accurately describe the data.

Line 222: ... and identify 3A6 IgG as a first-in-class antibody that appears to perform physical work. This statement is not correct, because the same mechanism has been proposed for HIV-1 MPER antibody 4E10. MPER is likely a dynamic structure and an extended MPER may be locked by 3A6. Do all GPs show the same height on the VLPs in the absence of 3A6?

Our response: Thank you for this insight. We agree that 3A6 (and other ebolavirus MPER-targeted mAbs) may lock this region in an extended position state. To better understand how 3A6 may access its epitope, we used AlphaFold 2 to model the glycoprotein, including MPER, interfacial region (IFR), and transmembrane domain (new Supplementary Fig. 6 and Fig. 4).. Notably, our modeling suggests the glycoprotein may sample both compacted and extended conformations of the MPER–IFR. The solution structure of the MPER transmembrane domain of the glycoprotein aligns well with the compact form, and the compact model of the full-length glycoprotein can be successfully docked to the KZ52-bound glycoprotein map but shows poor fit to the 3A6-bound glycoprotein map. Conversely, we find that the extended model of the glycoprotein fits well into the 3A6-bound map. . We hypothesize that, like other type I glycoproteins (such as those of HIV-1 and influenza A virus), EBOV glycoprotein “breathes” up and down, sampling the compact and extended states. We did not observe any binding of untransfected cells by 3A6 (now Supplementary Fig. 3); hence, 3A6 likely gains access to its epitope whilst the glycoprotein is in the extended state locks it in this conformation. We have deleted the reference to physical work or first-in-class.

Line 292: The manuscript contains no data that allows to claim that a portion of the 3A6 epitope is indeed embedded in the membrane.

Our response: We agree with the reviewer that our statement that the 3A6 epitope is embedded in the membrane was overly interpretive. As discussed above, we modeled the potential location of the 3A6 epitope in the context of full-length membrane-bound glycoprotein.

Supplementary Fig 4 has 5 panels A, B, C, D, E but the main text refers only to Supplementary Fig 4.

Our response: We have added the appropriate figure call-outs for what is now Supplementary Fig. 7.

Supplementary Fig 5 has two panels A and B, but the main text refers only to B.

Our response: Thank you for this attention to detail. We now reference both panels A and B in what is now Supplementary Fig 8.